# Assessing the quality of antenatal corticosteroids in low- and middle-income countries: A systematic review

Euodia Mosoro[1,2], Alyce N. Wilson[1,2], Caroline S. E. Homer[1,2], Joshua P. Vogel[1,2]*

1 Maternal, Child and Adolescent Health Program, Burnet Institute, Melbourne, Australia, 2 School of Population and Global Health, University of Melbourne, Melbourne, Australia

* joshua.vogel@burnet.edu.au

## Abstract

### Background

The World Health Organization (WHO) recommends the administration of intramuscular antenatal corticosteroids to women at risk of preterm birth to prevent preterm-associated neonatal mortality and morbidity. Poor quality medicines are a major problem for health services in low- and middle-income countries (LMICs), however the quality of antenatal corticosteroids is not well understood. We aimed to conduct a systematic review of available studies describing the quality of recommended injectable antenatal corticosteroids (dexamethasone or betamethasone) in LMICs.

### Methods

Structured search strategy was applied to six databases (MEDLINE, EMBASE, CINAHL, International Pharmaceutical Abstracts, Global Index Medicus, WHO Medicines Quality Database), without year or language restrictions. Any primary study reporting any medicine quality parameter (Active Pharmacological Ingredient, pH and sterility) for injectable dexamethasone or betamethasone was eligible. Two authors independently screened studies for eligibility, extracted data on included studies and applied Medicine Quality Assessment Reporting Guidelines tool to assess study quality. Results were reported narratively, stratified by country of manufacture, organisation type and level of care.

### Results

In total, 15,547 citations were screened with two eligible studies identified that focussed on dexamethasone quality (no studies of betamethasone were identified). One study included 19 samples from 9 LMICs, and the other included "less than 100 samples" from India. The prevalence of failed dexamethasone samples ranged from 3.14% to 32.2% due to inadequate Active Pharmacological Ingredient. A higher prevalence of failed dexamethasone samples were seen at the point of care and the public sector.

**Data Availability Statement:** All data extracted is provided in Supplementary Appendix.

**Funding:** The author(s) received no specific funding for this work.

**Competing interests:** The authors have declared that no competing interests exist.

## Conclusions

Poor quality maternal and newborn health medicines can endanger women and newborns. Though available evidence on antenatal corticosteroids quality in LMICs is limited, results suggested poor quality dexamethasone may be prevalent in some countries. More primary studies are required to confirm these findings and guide policymakers on procurement of good-quality maternal and newborn health medicines.

## Introduction

Preterm birth is defined as a baby born prior to 37 weeks' gestation [1]. Worldwide, an estimated 14.8 million babies are born preterm each year, with most of these babies (81%) being born in Asia and sub-Saharan Africa [2]. Preterm birth complications are the leading cause of death in children under five years of age accounting for an estimated 1.06 million deaths in 2015 [2, 3].

Antenatal corticosteroids are the mainstay of treatment for women at risk of preterm birth to prevent neonatal mortality and morbidity, particularly respiratory morbidities [4, 5]. In 2015, the World Health Organization (WHO) recommended the use of a single course of antenatal corticosteroids (24mg of dexamethasone or betamethasone, administered via intramuscular [IM] injections in divided doses) for women who are at risk of imminent preterm birth at 24 to 34 weeks' gestation [6]. Both dexamethasone and betamethasone were identified in the United Nations Commission on Life-Saving Commodities for Women and Children, and dexamethasone is listed on the WHO Model List of Essential Medicines [7, 8]. Antenatal corticosteroids, particularly dexamethasone, are widely available in LMICs and are commonly cited on national essential medicines lists [9, 10].

Quality assured medicines are imperative to ensure effective therapeutic outcomes. The quality of medicines are governed by a number of international standards and quality assurance programs, such as Good Manufacturing Practices, International Pharmacopeia 8[th] Edition and National Medicines Regulatory Authorities [11–13]. Substandard medicines are defined as genuine drugs that do not adhere to these standards, whereas counterfeit medicines are those that intentionally do not adhere to standards [13]. There is considerable evidence of substandard essential maternal and newborn health medicines in LMICs, such as oxytocin, misoprostol and magnesium sulfate [14–16]. Other reviews have demonstrated that substandard or counterfeit medicines are a widespread problem affecting antimalarials, TB treatments and antibiotics [17–19]. However, the quality of antenatal corticosteroids in LMICs has not been examined. We therefore aimed to identify and synthesise primary studies that examined the quality of antenatal corticosteroids—injectable (intramuscular, IM or intravenous, IV) dexamethasone or betamethasone in LMICs.

## Methods

We conducted a systematic review in accordance with a pre-specified protocol (PROSPERO CRD42020152107), in line with the Preferred reporting items for Systematic Reviews and Meta-analyses (PRISMA) guidance (see S1 Appendix for PRISMA Checklist) [20]. Search terms, databases and outcome measures were informed by a 2016 systematic review of oxytocin quality by Torloni et al. [14]. Specifically, we adopted their review outcome Active Pharmacological Ingredient as one of our outcomes; used the same quality assessment tool and score

cut-offs for study quality; and we developed our search strategy informed by search terms used by Torloni et al.

The quality of a medicine can be assessed based on multiple parameters, include the macroscopic appearance, extractable volume, pH range, proportion of active pharmacological ingredient (API), sterility, solubility, quality of the excipients and proportion of acceptable contaminants [21]. Furthermore, correct labelling and maintenance of storage conditions are essential to maintain quality during transport to health facilities [11, 22]. The acceptable range for these parameters is typically defined by the manufacturer and the relevant national or international pharmacopeia. Quality parameters for API, pH and sterility have been defined by the International Pharmacopeia (Box 1) and these three attributes are the most salient in assessing medicine quality [23, 24]. Injectable dexamethasone and betamethasone must be stored in a cool, dry place, away from light, kept below 25˚C and not frozen; storage temperatures may differ slightly with different manufacturers [25]. US and British Pharmacopeia use the same parameters [21, 26]. We considered the API failure rate, sterility and pH of injectable dexamethasone and betamethasone (as reported by the authors) as the main review outcomes. By API failure, we mean samples that did not meet the API quality parameters designated by US and British Pharmacopeia, in terms of API concentration or pH [21, 26]. For dexamethasone, API failure included API concentration <90% ("low fail"), API >110% ("high fail") or a pH less than 7.5 or greater than 8.5. For betamethasone, API failure included API <96% ("low fail"), API >104% ("high fail") or a pH less than 7.0 or greater than 8.5.

## Eligibility criteria

Eligible studies and reports were those describing the quality of injectable (IM or IV) dexamethasone sodium phosphate, betamethasone phosphate or betamethasone acetate for use in preterm birth in LMICs. Specifically, any primary study (regardless of design, whether observational or interventional) that reported API, sterility or pH using valid laboratory methods was considered eligible. LMICs were defined using the World Bank classification [27]. Studies were included irrespective of sample size, year of sample collection, date of publication, language, or whether samples were collected from public or private supply chains. Those studies that solely validated high-performance liquid chromatography (HPLC) techniques for the quantification of dexamethasone or betamethasone concentration were excluded, as these are performed using samples of known concentrations. We also excluded studies simulating storage conditions to examine stability. Commentary papers, reviews and editorials without primary data were excluded.

## Information sources and search strategy

The following electronic databases were searched up to July 2019 without language or date restrictions: MEDLINE, EMBASE, International Pharmaceutical Abstracts (IPha), CINAHL, Global Indicus Medicus and the WHO Medicines Quality Database. Search terms include synonyms of dexamethasone, betamethasone, corticosteroid and preterm birth (see S2 Appendix for search strategy). We contacted key maternal and newborn health and medicine quality stakeholders, including the Reproductive Health Supplies Coalition and Maternal Health Supplies Caucus [28]. We also identified 45 international antenatal corticosteroid manufacturers and contacted them via email to request any eligible reports.

## Study selection, data collection process and quality assessment

All citations were uploaded into EndNote and duplicates removed, which then were screened using Covidence [29]. All citations were initially assessed on the basis of title and abstract. Full text review was conducted for potentially eligible citations. All screening was conducted by

## Box 1. Essentials medicines and dosage

| Essential Medicine and Dosage | API | pH | Sterility |
|---|---|---|---|
| Dexamethasone sodium phosphate ($C_{22}H_{30}FO_8P$) (4mg/mL) in 1 mL ampoule. | 90% - 110% | | |
| Betamethasone ($C_{22}H_{29}FO_5$)<br>• 5.7mg/mL (3mg/mL as betamethasone sodium phosphate + 2.7mg/mL as betamethasone acetate) in 1mL ampoules<br>• 4mg/mL as betamethasone sodium phosphate in 1mL or 2mL ampoules. | 96.0% - 104% | 7.0 to 8.5 | No micro-organisms |

two independent reviewers (EM and JV) with conflicts resolved by a third reviewer (CH). Where necessary, Google Translate was used to assess eligibility for non-English articles. Data were extracted from included studies using a pre-designed Excel spreadsheet. Data extracted included: study type, medicine and sample characteristics; quality testing parameters; level of care sample was collected from, pharmacopeia used; laboratory tests used for quality assessment, as well as the outcomes API, pH, sterility, storage conditions, temperature and humidity. Included studies were assessed for methodological quality using the 12-domain Medicine Quality Assessment Reporting Guidelines (MEDQUARG) tool (see S3 Appendix) [24]. Studies of good methodological quality were defined as those with a MEDQUARG score ≥6, and those with score <6 as being of low quality.

### Data synthesis

The proportion of failed samples (whether low-fail or high-fail) were reported narratively, as described by the authors. We planned to conduct meta-analysis of outcome data, as well as sensitivity analyses by manufacturer type, level of care and country income level, however the limited data available were too heterogeneous to do so. Hence this review was confined to descriptive analysis only.

### Results

A total of 18,501 citations were identified, of which 2954 were duplicate records (Fig 1). Of the 15 467 unique citations screened for eligibility, 62 were subjected to full text review and two studies met the inclusion criteria and were included [23, 30].

### Characteristics of included studies

The two included reports used cross-sectional designs, and examined the quality of dexamethasone and betamethasone in addition to other essential medicines (Table 1). One was a 2015

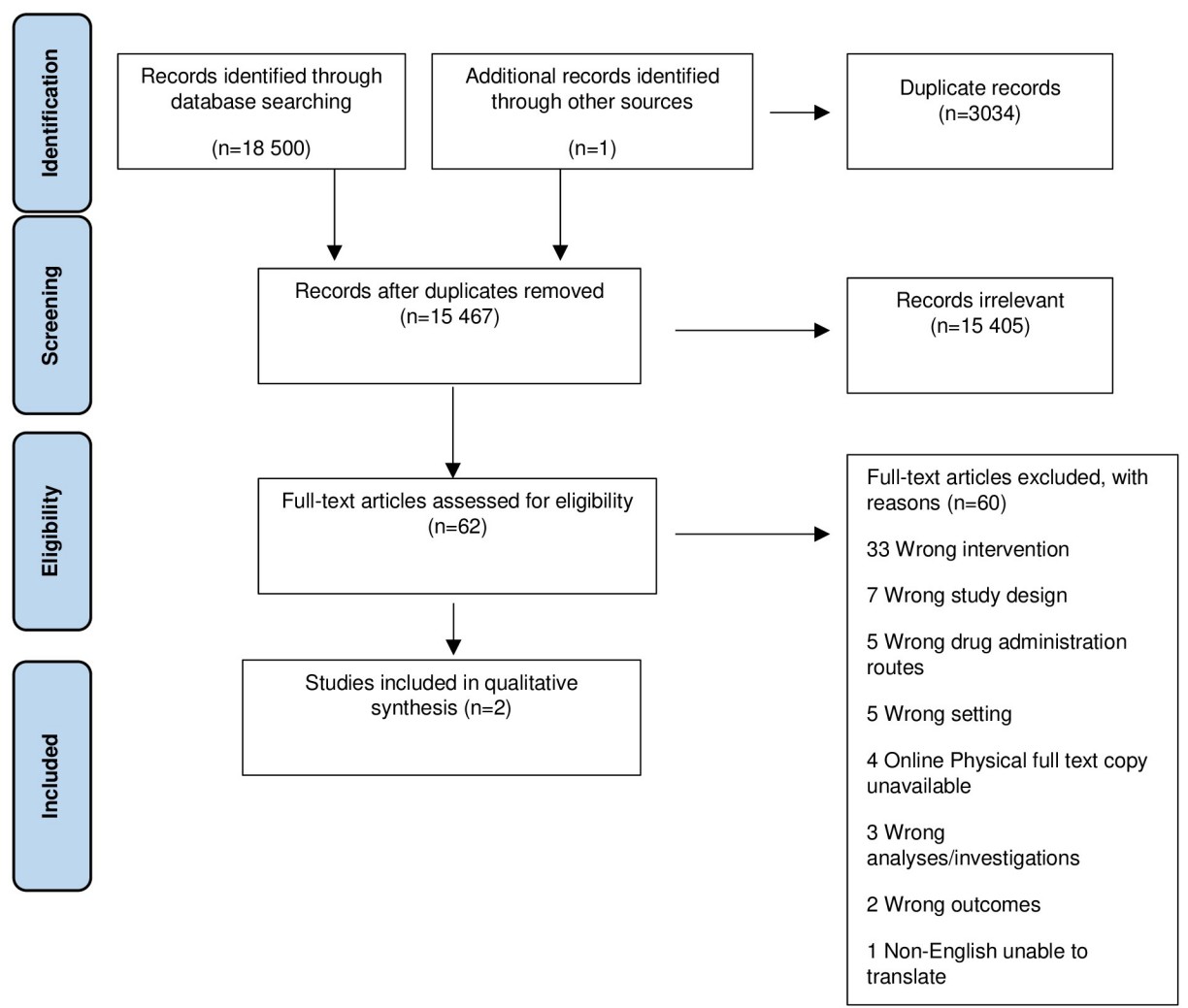

**Fig 1. PRISMA flow diagram.**

report by the United Nations Commission on Life Saving Commodities for Women and Children, and the other was a 2014 national quality medicines survey completed by the Ministry of Health and Welfare, Government of India [23, 30]. The UN Commission multi-country survey reported on 19 samples of dexamethasone from 9 countries (India, Kenya, Madagascar, Nepal, Nigeria, Tanzania, Tajikistan, Uganda, Vietnam, Zimbabwe) (Table 1) [23]. The Government of India report included "less than 100 samples" of dexamethasone only [30]. Methodological quality based on MEDQUARG assessment was good (8 of 12) (Table 1, S3 Appendix). Across

**Table 1. Injectable dexamethasone and betamethasone quality parameters, based on International Pharmacopeia [25, 44] and US Pharmacopeia [45].**

| Medicine and dosage | API | pH | Sterility |
|---|---|---|---|
| Dexamethasone sodium phosphate | 90%–110% | 7.0 to 8.5 | No micro-organisms |
| Betamethasone sodium phosphate | 96%–104% | 7.0 to 8.5 | No micro-organisms |
| Betamethasone acetate | 97–103% | 6.8 to 7.2 | No micro-organisms |

both reports, all samples were collected in 2013 from either distributors, central medical stores, wholesalers or hospitals (no samples were from drug stores, informal sellers or online sales). Samples for the study in India were collected as part of a nationwide medicine quality study that covered 224 drug molecules. The samples were collected from across 654 different districts in India; the districts where dexamethasone was sample were not specified.

The authors of the UN Commission multi-country survey reported that 9 samples of dexamethasone were manufactured in China, 6 samples from Indian manufacturers, 3 samples from Vietnamese manufacturers and 1 from a Russian manufacturer (S4 Appendix) [23]. The Government of India report did not disaggregate by manufacturer, reporting that "less than 50" samples were from private and "less than 50" from public sectors. The exact API values were reported for all samples in the UN Commission study (S4 Appendix) and ranged from 64.1% to 10.5.1%, however exact API values were not available from the Government of India study.

## Main outcomes

Both reports provided outcome data on appearance, assay, pH and extractable volume of dexamethasone (Table 1). The UN Commission multi-country survey also provided data on the presence of free dexamethasone in samples. While the two reports assessed samples against different pharmacopeia (British and Indian), both are aligned with International Pharmacopeia standards on dexamethasone. Overall, the prevalence of low fails was between 3.14% to 32.2% (Table 1).

The UN Commission multi-country survey assayed a total of 19 dexamethasone samples and reported a low fail prevalence of 32.2% (6/19) (Table 2). In total, 50% of failed samples were from African countries and the remainder from South-East Asian countries. The countries that reported one failed sample of dexamethasone were Kenya, Madagascar, Nigeria, Tajikistan, Vietnam and Nepal. The failed samples included three dexamethasone ampoules that were considered failures due to the presence of free dexamethasone (S4 Appendix). The Government of India survey reported failed samples of dexamethasone separately for private and public sector. The prevalence of low fails was 3.14% for private sector and 20.2% for public sector.

Table 3 presents results stratified by country of manufacturer (national or international), organisation type (public or private) and level of care (central or point of care), however the absolute numbers of samples for these comparisons is quite low. In the UN Commission survey the proportion of failed dexamethasone samples was slightly higher for national manufacturers (33%) than international manufacturers (31%), point of care samples had more failures (33%) than central level samples (31%), and private sector samples (36%) had higher failure rates than public sector (20%).

## Discussion

### Main findings

This systematic review identified available evidence on the quality of dexamethasone or betamethasone in LMICs, a critical intervention in preterm birth management. We identified only two good-quality studies (19 samples and "less than 100" samples) of dexamethasone; no studies reported on betamethasone samples. While sample size was limited, inadequate quality dexamethasone was identified in African and South-East Asian countries, mainly due to inadequate concentration of the active pharmacological ingredient. Given the limited number of dexamethasone samples available and the lack of quality data on betamethasone, these findings

**Table 2. Characteristics and quality parameters of included studies.**

| Reference | Country | Year Sample | Study Quality Grade | Total number of samples assayed | Manufacturer Country | Level of care | Prevalence of Low Fails | Stated problem | Percent API fails >95–110% |
|---|---|---|---|---|---|---|---|---|---|
| United Nations Commission of Life Saving Commodities | Kenya, Madagascar, Nepal, Nigeria, Tanzania, Tajikistan, Uganda, Vietnam, Zimbabwe | 2013 | 8 | | China (9) | Wholesale Private | 32.2% (6) | Inadequate API | 32.2% (6) |
| | | | | 19 | India (6) | Distributor Private | | Presence of free dexamethasone. | |
| | | | | | Russia (1) | Public Treatment Centre | | | |
| | | | | | Vietnam (3) | Public Hospital | | | |
| | | | | | | Importer Distributor | | | |
| Ministry of Health and Welfare Government of India | India | 2013 | 8 | <50 | NI | Wholesale Private | 3.14%* | Inadequate API | 3.14%* |
| | | | | <50 | NI | Public Treatment Centre & Hospital | 20.2% * | Inadequate API | 20.2% * |

NI = No information.

should be interpreted with some caution until further research of antenatal corticosteroid quality becomes available.

## Strengths and limitations

To our knowledge, this systematic review is the first that has examined the quality of injectable dexamethasone and betamethasone in LMICs. We used a broad and systematic approach to identify eligible studies, including contacting key stakeholders and manufacturers. One limitation of this review is the possibility of publication bias—countries and manufacturers may be disinclined to publicly release studies indicative of poor medicine quality. Despite our efforts to contact individual stakeholders and manufacturers directly, it is possible that national regulatory authorities, manufacturers or other organisations involved in drug procurement have more extensive information on the quality of common medicines such as dexamethasone and betamethasone. The two included studies did not provide data on all quality parameters of interest to this review, hence we were unable to assess other medicine quality parameters such as pH, sterility and storage conditions. While it is possible that new studies may have emerged since our search was conducted, we consider the likelihood of this to be low considering how few studies were identified. Nevertheless, we plan to update this review in the future to identify any emerging evidence.

## Interpretation

The impact of poor-quality medicines has profound public health implications. Low concentration of API in dexamethasone samples could potentially lead to subtherapeutic dosing of women at risk of preterm birth. Antenatal corticosteroids have been shown to confer a 31% reduction in the relative risk of neonatal death, as well as substantial reductions in the risk of severe respiratory distress syndrome, intraventricular haemorrhage, necrotising enterocolitis and need for mechanical ventilation [5]. It is possible that these important health benefits

**Table 3. Proportion of failed dexamethasone samples according to country of manufacturer, organisation type and level of care.**

| Study | Country | All Samples | | | Manufacturers | | | | | | Organisation Type | | | | | | Level of care | | | | | |
|---|---|---|---|---|---|---|---|---|---|---|---|---|---|---|---|---|---|---|---|---|---|---|
| | | Total | Failed Sample | | International | | | National | | | Public | | | Private | | | Central | | | Point of Care | | |
| | | | | | Total | | | Total | | | Total | | | Total | | | Total | | | Total | | |
| | | | n | % | n | n | % | n | n | % | n | n | % | n | n | % | n | n | % | n | n | % |
| United Nations Commission of Life Saving Commodities. | Kenya, Madagascar, Nepal, Nigeria, Tajikistan, Tanzania, Uganda, Vietnam, Zimbabwe | 19 | 6 | 33 | 16 | 5 | 31.2 | 3 | 1 | 33.3 | 5 | 1 | 20 | 14 | 5 | 35.7 | 16 | 5 | 31 | 3 | 1 | 33.3 |
| Ministry of Health and Welfare Government of India (MoHW) | India | <100 | NR | | NR | | | | | | <50 | NI | 20.2 | <50 | NR | 3.4 | NR | | | | | |

NR = not reported.

NI = no information reported by authors.

cannot be achieved (or only partially achieved) with low API dexamethasone. Furthermore, poor quality medicines may cause harm and decrease the confidence of health workers in the effectiveness of treatments [31, 32]. There are also economic losses to health systems and out-of-pocket costs borne by patients [33].

Concerns regarding medicine quality are not confined to corticosteroids—other maternal health medicines have had similar problems identified. A 2020 systematic review by Torloni et al. [34] identified 34 studies on maternal health medicine quality in low- and middle-income countries, identifying concerningly high failure rates for the uterotonics ergometrine (75.4%), oxytocin (39.7%), misoprostol (38.7%), as well as injectable antibiotics (13.4%) and magnesium sulphate (3.4%). A high prevalence of inadequate API has also been identified in antimalarial medicines (42–48%) and TB treatments (10%) [17, 18, 35]. A 2018 systematic review by Ozawa et al. [33] examined 265 studies of medicine quality, estimating the overall prevalence of substandard and counterfeit medicines in LMICs as 13.6%. While the evidence of poor-quality dexamethasone in LMICs is a concern, it is of even greater concern that so few studies of antenatal corticosteroid quality were identified. The findings of previous systematic reviews of medicine quality emphasise there is significantly more studies on the quality of medicines other than antenatal corticosteroids, despite their critical importance in preterm birth management.

In 2008, Caudron et al. [31] identified three factors as likely causes for the existence of substandard medicines. Firstly, the poor compliance to Good Manufacturing Practices by manufacturers. For example, Nebot et al. [36] found that pharmaceutical distributors do not apply or adhere to stringent standards of quality when supplying to sub-Saharan African markets, and in turn these African countries had weak regulatory oversight to enforce and ensure accountability of the international pharmaceutical distributors. Second and third, the limited pharmacovigilance capacity in LMICs, as well as the limited financial and human resource capacity of in-country national medicines regulatory authorities within LMICs can further contribute to substandard medicines. A 2010 review of medicines regulatory systems in 26 sub-Saharan African countries identified that 54% of countries did not have quality monitoring and surveillance programs, sustainable funding and technical capacity of staff to oversee these initiatives [37].

There are a range of international health initiatives that focus on capacity building of quality assurance programs of essential medicines in LMICs, such as the WHO Model Quality Assurance System to standardise procedures of procurement agencies, and the WHO Global Benchmarking tool to evaluate national medicines regulatory systems [33, 37]. Other global health initiatives, such as the WHO Programme for International Drug Monitoring towards Global Pharmacovigilance and WHO Quality Medicines Database are also aimed at improving quality-assured medicines [38, 39]. In the field of maternal and newborn health, initiatives such as the Reproductive Health Supplies Coalition (RHSC) focus on improving the availability of specific medicines (such as oxytocin, magnesium sulphate, misoprostol) [28]. One of the RHSC objectives is to work with generic manufacturers towards WHO prequalification to ensure the quality, efficacy and safety of supplied medicines [40] WHO prequalification may be a route to ensure better quality antenatal corticosteroids—at present, only one manufacturer of dexamethasone has been prequalified [41]. However, prequalification alone will not address the complex issue of substandard medicines in LMICs [42].

A key recommendation from this review is that more primary studies of injectable dexamethasone or betamethasone are required. Ideally, primary studies would have substantive sample sizes, use random sampling (in accordance with WHO guidelines on surveys of medicine quality) consider a range of countries and ensure samples are available from different points of the supply and transport chain [43]. Considering the concerning findings of this review, such studies are an urgent next step in ensuring that preterm birth management in LMICs is as safe and effective as possible.

## Conclusion

The presence of poor quality maternal and newborn health essential medicines in LMIC present health risks to women and newborns. This systematic review identified that poor quality dexamethasone is present in several LMICs, raising concerns as to whether antenatal corticosteroids can be used safely and effectively in these countries. Further primary studies of dexamethasone or betamethasone are required to better understand the extent and consequences of these quality issues, and how they can be addressed.

## Supporting information

**S1 Appendix. PRISMA checklist.**
(DOC)

**S2 Appendix. Search strategy.**
(DOCX)

**S3 Appendix. Quality assessment of studies using MEDQUARG tool [24].**
(DOCX)

**S4 Appendix. Reported quality parameters for all available data.**
(DOCX)

## Acknowledgments

The authors wish to thank Pete Lambert and A. Metin Gülmezoglu for their assistance.

## Author Contributions

**Conceptualization:** Euodia Mosoro.

**Formal analysis:** Euodia Mosoro.

**Investigation:** Euodia Mosoro.

**Methodology:** Euodia Mosoro, Alyce N. Wilson, Caroline S. E. Homer, Joshua P. Vogel.

**Project administration:** Caroline S. E. Homer.

**Supervision:** Alyce N. Wilson, Caroline S. E. Homer, Joshua P. Vogel.

**Writing – original draft:** Euodia Mosoro.

**Writing – review & editing:** Euodia Mosoro, Alyce N. Wilson, Caroline S. E. Homer, Joshua P. Vogel.

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
