## [Decision Letter · Decision Letter 0]

24 Sep 2020

PONE-D-20-24549

Assessing the quality of antenatal corticosteroids in low- and middle-income countries: a systematic review.

PLOS ONE

Dear Dr. Vogel,

Thank you for submitting your manuscript to PLOS ONE. After careful consideration, we feel that it has merit but does not fully meet PLOS ONE’s publication criteria as it currently stands. Therefore, we invite you to submit a revised version of the manuscript that addresses the points raised during the review process.

We look forward to receiving your revised manuscript.

Kind regards,

Chaisiri Angkurawaranon

Academic Editor

PLOS ONE

Journal Requirements:

2. Thank you for stating the following beneath the Acknowledgments Section of your manuscript:

'FUNDING

Maternal, Child and Adolescent Health Program, Burnet Institute, Melbourne, Australia'

'The author(s) received no specific funding for this work.'

3. Please include a new copy of Table 3 in your manuscript; the current table is difficult to read.

Please follow the link for more information: https://blogs.plos.org/plos/2019/06/looking-good-tips-for-creating-your-plos-figures-graphics/

Reviewers' comments:

Reviewer's Responses to Questions

**Comments to the Author**

1. Is the manuscript technically sound, and do the data support the conclusions?

Reviewer #1: Partly

Reviewer #2: Partly

2. Has the statistical analysis been performed appropriately and rigorously? 

Reviewer #1: N/A

Reviewer #2: Yes

3. Have the authors made all data underlying the findings in their manuscript fully available?

Reviewer #1: Yes

Reviewer #2: No

4. Is the manuscript presented in an intelligible fashion and written in standard English?

Reviewer #1: Yes

Reviewer #2: Yes

5. Review Comments to the Author

Reviewer #1: PONE-D-20-24549 Feedback

Assessing the quality of antenatal corticosteroids in low- and middle-income countries: a systematic review.

PLOS ONE

General comments:

The authors have conducted research on an original and interesting topic. The methodology of the systematic review is acceptable. Suggestions to consider for improving the quality and reporting of the study are outlined, below.

In the discussion, the authors have demonstrated knowledge of the medicine quality, in general. However, there seem to be few paragraphs of the discussion linked directly to the objective of the systematic review – assessing the quality of antenatal corticosteroids in low- and middle-income countries.

Detailed comments

Introduction

In 2015, the World Health Organization (WHO) … for women wo are at risk of imminent preterm birth at 24 to 34 weeks’

gestation.(6)

Kindly correct typo ‘wo’ in the statement

We therefore aimed to identify and synthesise primary studies that examined the quality of antenatal corticosteroids (injectable dexamethasone or betamethasone) in LMICs.

Please could the authors clarify here whether the review focused on injectable corticosteroids (IM and IV or either) for all indications or for antenatal use, only

It will be useful to provide the indication of the intervention, i.e. management of preterm birth (as stated in ‘Study identification’)

Methods

General comments

For clarity, kindly re-arrange text in this section to follow the order recommended by the PRIMSA guidance, e.g. Study identification (eligibility criteria), Information sources, Search etc.

Kindly ensure referencing style is consistent, throughout

Methods for undertaking the systematic review refers to 2 citations ([20] and [32])

Please clarify how [32] informed the methods

Please check the referencing style of [32]

Search terms, databases and outcome measures were informed by a 2016 systematic review of oxytocin quality by Torloni et al.

For clarity, could the authors kindly state how the Torloni review informed their methodology? E.g., search terms; quality assessment; choice of analysis etc.

Study identification

It will be helpful to elucidate the study designs of interest.

Results indicate data from LICs and LMICs. Please clarify whether LICs are a subgroup of LMICs. If this is not the case, kindly address this as appropriate to avoid concerns with reviewer bias

Please check this sentence for comprehension, ‘Specifically, any primary study that reported API, sterility or pH using valid laboratory methods.’

Search strategy

The authors indicated that electronic searches were conducted in July 2019. Is it likely that recent relevant studies could have been missed?

Quality Assessment

The authors defined the criterion for good methodological quality. Please can you add the rating for remaining categories?

Data analysis

Kindly explain the term ‘failed samples’ as best as possible

Please could the authors indicate how data synthesis was undertaken?

Results

The authors of the UN Commission multi-country survey reported that dexamethasone was

manufactured in China (9 manufacturers), India (6), Russia (1) and Vietnam (3)

Please clarify what (6), (1) and (3) refer to , here

Primary and secondary outcomes

This is first mention of primary and secondary outcomes (including appearance and extractable volume).

Please justify these outcomes

To provide clarity, authors should kindly consider providing more detail for ‘failed’ or ‘low fail’ samples with respect to quality components examined (e.g. API, pH, sterility) and relevant assessment rating.

TABLES

Table 2 shows ‘Appearance, Assay, pH, Extractable Volume, Free Dexamethasone’ as Tests Performed. This is a bit confusing.

The table below Table 3 (page 19 of 23) has no title. It is unclear whether parts of the table are missing.

FIGURE

Figure 1: Kindly revise title to PRISMA flow diagram or Study selection

Numbers in the box – Full-text articles excluded with reasons up to 56 not 60. Please check

Discussion

Author present their findings in the light of previous relevant literature, perceptively and in detail but there is limited discussion of their findings (quality of studies, methodological differences etc.).

In the recommendations, authors missed an opportunity to highlight and link future research needs and the dearth of evidence relating to various quality components of injectable drugs.

Reviewer #2: General comments

Even though the quality of the pharmacologic preparations of antenatal corticosteroids vary, a meta-analysis on the subject has not been conducted (to my knowledge). This systematic review of available studies is designed to describe the quality of available injectable antenatal corticosteroids (dexamethasone or betamethasone) in LMICs. The review revealed that the prevalence of failed dexamethasone samples ranged from 3.14% to 32.2% due to inadequate active pharmacological ingredient (API). This is an important finding in view of the dependence of LMICs on sourcing high quality antenatal steroids (ANS) even though it is based on limited data.

More emphasis should be placed on the exact values of API found in the two studies rather than relying on agreed upon but arbitrary cut-off values as dexamethasone results in a wide dose-effect response..

Introduction

The authors have a good rationale for studying the quality of ANS as there is considerable evidence of substandard essential medicines in LMICs such as anti-malarials, TB treatments, antibiotics, oxytocin, misoprostol and magnesium sulfate.

Methods

Excellent methods were used which included a pre-specified protocol (PROSPERO CRD42020152107) and the Preferred Reporting Items for Systematic Reviews and Meta-analyses (PRISMA) guidance.

Results

The data are very limited with only 19 samples in the UN study and “less than 100” in the Indian study. This is an important limitation that should be highlighted in the Abstract (which otherwise implies that the results are based on good data).

Even though less than 100 samples were tested in the study from India, it is stated that samples from India were collected from 654 different districts where dexamethasone was available. Collecting samples from 654 different districts is inconsistent with only testing less than 100. Please clarify.

Please clarify the statement “The countries that reported one failed sample of dexamethasone were Kenya, Madagascar, Nigeria, Tajikistan and Vietnam with Nepal having two failed samples” as this would total 7 samples but the text states that 6 samples failed.

Would it be possible to report the exact API values instead on relying only in reaching a narrow range of API? These data would be most useful for clinicians as BMI varies by over 2 fold.

Discussion

It is stated “We identified only two studies of 119 samples of dexamethasone…) but they authors have stated that the number of samples was 19 in one study and “less than 100” in the other. Please clarify.

Dexamethasone dosing is not exact so it is important to put into perspective the range of API found to determine the potential clinical significance of the findings.

6. PLOS authors have the option to publish the peer review history of their article (what does this mean?). If published, this will include your full peer review and any attached files.

Reviewer #1: No

Reviewer #2: No

---

## [Author Response · Author response to Decision Letter 0]

3 Nov 2020

Dear Editor,

Many thanks for the helpful comments on our paper. We have submitted a revised version, and below is a point-by-point response to the issues raised.

Many thanks

Euodia Mosoro and Joshua Vogel on behalf of the authors

General comments:

The authors have conducted research on an original and interesting topic. The methodology of the systematic review is acceptable. Suggestions to consider for improving the quality and reporting of the study are outlined, below.

In the discussion, the authors have demonstrated knowledge of the medicine quality, in general. However, there seem to be few paragraphs of the discussion linked directly to the objective of the systematic review – assessing the quality of antenatal corticosteroids in low- and middle-income countries.

Response: Thanks, please see our response below to the relevant detailed comments blow.

Detailed comments:

Introduction

1. In 2015, the World Health Organization (WHO) … for women wo are at risk of imminent preterm birth at 24 to 34 weeks’ gestation.(6) Kindly correct typo ‘wo’ in the statement

Response: Done

2. We therefore aimed to identify and synthesise primary studies that examined the quality of antenatal corticosteroids (injectable dexamethasone or betamethasone) in LMICs Please could the authors clarify here whether the review focused on injectable corticosteroids (IM and IV or either) for all indications or for antenatal use, only It will be useful to provide the indication of the intervention, i.e. management of preterm birth (as stated in ‘Study identification’)

Response: For the purpose of the systematic review we focused on injectable corticosteroids (either IV or IM) that were intended for antenatal use, specifically for the management of preterm birth. We have clarified this in the final para of Background, and also under study identification.

3. Methods General comments. For clarity, kindly re-arrange text in this section to follow the order recommended by the PRIMSA guidance, e.g. Study identification (eligibility criteria), Information sources, Search etc. Kindly ensure referencing style is consistent, throughout. 

Response: We have reworded subheadings to better reflect the PRISMA checklist. We have corrected references throughout.

4. Methods for undertaking the systematic review refers to 2 citations ([20] and [32]) Please clarify how [32] informed the methods. Please check the referencing style of [32].

Response: Apologies, there was a referencing software error – this reference was meant to be a single reference, to Torloni 2016 – reference now corrected.

The Torloni review focused on the quality of oxytocin in low to middle income countries and informed the methodology of this systematic review. We have added the following sentence to the first paragraph of Methods:

“Specifically, we adopted their review outcome Active Pharmacological Ingredient as one of our outcomes; used the same quality assessment tool and score cut-offs for study quality; and we developed our search strategy informed by search terms used by Torloni et al.”

5. Search terms, databases and outcome measures were informed by a 2016 systematic review of oxytocin quality by Torloni et al. For clarity, could the authors kindly state how the Torloni review informed their methodology? E.g., search terms; quality assessment; choice of analysis etc.

Response: Please see our response to #4 above.

6. Study identification It will be helpful to elucidate the study designs of interest. 

Response: We have added detsail under “Eligibility Criteria” to clarify:

“Eligible studies and reports were those describing the quality of injectable (IM or IV) dexamethasone sodium phosphate, betamethasone phosphate or betamethasone acetate for use in preterm birth in LMICs. Specifically, any primary study (regardless of design, whether observational or interventional) that reported API, sterility or pH using valid laboratory methods was considered eligible. LMICs were defined using the World Bank classification.(27)”

7. Results indicate data from LICs and LMICs. Please clarify whether LICs are a subgroup of LMICs. If this is not the case, kindly address this as appropriate to avoid concerns with reviewer bias

Response: In considering studies, we considered any low to middle income country based on the World Bank definition. (see revision included in response to point 6 above). We have clarified Table 2 to remove LICs and avoid definitional confusion. 

8. Please check this sentence for comprehension, ‘Specifically, any primary study that reported API, sterility or pH using valid laboratory methods.’

Response: Thanks, we have edited this sentence for clarity: “Specifically, any primary study that reported API, sterility or pH using valid laboratory methods was considered eligible.”

9. Search strategy. The authors indicated that electronic searches were conducted in July 2019. Is it likely that recent relevant studies could have been missed?

Response: Thanks for this point. We acknowledge search was conducted in July 2019, though we consider the likelihood of more recent studies to be quite low, considering that no studies were identified in the published literature, and the 2 identified reports were from grey literature.

For clarity, we have added the following sentence to the Discussion:

“While it is possible that new studies may have emerged since our search was conducted, we consider the likelihood of this to be low considering how few studies were identified. Nevertheless, we plan to update this review in the future to identify any emerging evidence.”

10. Quality Assessment. The authors defined the criterion for good methodological quality. Please can you add the rating for remaining categories?

Response: Thanks for this point. We adopted the approached used by Torloni et al, and have reworded the relevant sentence for clarity: “Studies of good methodological quality were defined as those with a MEDQUARG score ≥6, and those with score <6 as being of low quality.”

11. Data analysis. Kindly explain the term ‘failed samples’ as best as possible

Response: We have edited para 2 of Methods section for clarifying this definition. Now reads:

“By API failure, we mean samples that did not meet the API quality parameters designated by US and British Pharmacopeia, in terms of API concentration or pH.(25, 26) For dexamethasone, API failure included API concentration <90% (“low fail”) , API >110% (“high fail”) or a pH less than 7.5 or greater than 8.5. For betamethasone, API failure included API <96% (“low fail”), API >104% (“high fail”) or a pH less than 7.0 or greater than 8.5.”

12. Please could the authors indicate how data synthesis was undertaken?

Response: We have clarified under “Data Synthesis” that available data were reported descriptively.

“The proportion of failed samples (whether low-fail or high-fail) were reported narratively, as described by the authors. We planned to conduct meta-analysis of outcome data, as well as sensitivity analyses by manufacturer type, level of care and country income level, however the limited data available were too heterogeneous to do so. Hence this review was confined to descriptive analysis only.”

13. Results. The authors of the UN Commission multi-country survey reported that dexamethasone was manufactured in China (9 manufacturers), India (6), Russia (1) and Vietnam (3) Please clarify what (6), (1) and (3) refer to here.

Response: We have revised wording for clarity: “The authors of the UN Commission multi-country survey reported that 9 samples of dexamethasone were manufactured in China, 6 samples from Indian manufacturers, 3 samples from Vietnamese manufacturers and 1 from a Russian manufacturer (Appendix S4).(23) The Government of India report did not disaggregate by manufacturer, reporting that “less than 50” samples were from private and “less than 50” from public sectors.”

14. Primary and secondary outcome. This is first mention of primary and secondary outcomes (including appearance and extractable volume). Please justify these outcomes

Response: Thanks, good point. We have revised this subheading to “main outcomes”, which is how they have been described in the Methods section.

15. To provide clarity, authors should kindly consider providing more detail for ‘failed’ or ‘low fail’ samples with respect to quality components examined (e.g. API, pH, sterility) and relevant assessment rating.

Response: Please see response above to #11.

16. Table 2 shows ‘Appearance, Assay, pH, Extractable Volume, Free Dexamethasone’ as Tests Performed. This is a bit confusing.

Response: Thanks, we agree this column in Table 2 is unnecessary and distracts from the main messages. We have removed it. 

17. The table below Table 3 (page 19 of 23) has no title. It is unclear whether parts of the table are missing.

Response: Thanks for comment re: Table 3 – a slight formatting error. Title now visible.

18. Figure 1: Kindly revise title to PRISMA flow diagram or Study selection

Response: Title has been revised. 

19. Numbers in the box – Full-text articles excluded with reasons up to 56 not 60. Please check

Response: Re: excluded articles - Thanks – a transcription error. We have corrected it. 

20. Discussion - Author present their findings in the light of previous relevant literature, perceptively and in detail but there is limited discussion of their findings (quality of studies, methodological differences etc.). In the recommendations, authors missed an opportunity to highlight and link future research needs and the dearth of evidence relating to various quality components of injectable drugs.

Response: Thanks, this is a really helpful observation. We have added the following sentences to para 4 of the Discussion to emphasise this point:

“While the evidence of poor-quality dexamethasone in LMICs is of concern, it is of even greater concern that so few studies of antenatal corticosteroid quality were identified. The findings of previous systematic reviews of medicine quality emphasise there is significantly more studies on the quality of medicines other than antenatal corticosteroids, despite their critical importance in preterm birth management.”

We have also modified the final para of Discussion on research recommendations:

“A key recommendation from this review is that more primary studies of injectable dexamethasone or betamethasone are required. Ideally, primary studies would have substantive sample sizes, use random sampling (in accordance with WHO guidelines on surveys of medicine quality) consider a range of countries and ensure samples are available from different points of the supply and transport chain.(42) Considering the concerning findings of this review, such studies are an urgent next step in ensuring that preterm birth management in LMICs is as safe and effective as possible.”

Reviewer #2: General comments

21. Even though the quality of the pharmacologic preparations of antenatal corticosteroids vary, a meta-analysis on the subject has not been conducted (to my knowledge). This systematic review of available studies is designed to describe the quality of available injectable antenatal corticosteroids (dexamethasone or betamethasone) in LMICs. The review revealed that the prevalence of failed dexamethasone samples ranged from 3.14% to 32.2% due to inadequate active pharmacological ingredient (API). This is an important finding in view of the dependence of LMICs on sourcing high quality antenatal steroids (ANS) even though it is based on limited data.

Response: We agree with this summary, thanks.

22. More emphasis should be placed on the exact values of API found in the two studies rather than relying on agreed upon but arbitrary cut-off values as dexamethasone results in a wide dose-effect response.

Response: Thanks for this comment. We adopted the cut-off values specified by the US and British Pharmacopeia to determine the quality of dexamethasone or betamethasone. We consider it important to emphasise those thresholds given their importance internationally in assessing dexamethasone or betamethasone quality in a standardised manner. Furthermore, while the United Nations Commission on Life Saving Commodities report described exact API values, the second study (Ministry of Welfare India), did not. We have provided the exact API values from the UN Commission report in Appendix S4, and added the following sentence to the Results:

“The exact API values were reported for all samples in the UN Commission study (Appendix S4) and ranged from 64.1% to 10.5.1%, however exact API values were not available from the Government of India study.

if more primary studies and surveys were completed to examine the quality of dexamethasone and betamethasone examining the dose-effect response would be a valuable amendment to the methodology.

23. Introduction. The authors have a good rationale for studying the quality of ANS as there is considerable evidence of substandard essential medicines in LMICs such as anti-malarials, TB treatments, antibiotics, oxytocin, misoprostol and magnesium sulfate.

Response: Thanks, we agree. The issue of poor-quality medicines is concerningly pervasive. 

24. Methods .Excellent methods were used which included a pre-specified protocol (PROSPERO CRD42020152107) and the Preferred Reporting Items for Systematic Reviews and Meta-analyses (PRISMA) guidance.

Response: thanks

25. Results. The data are very limited with only 19 samples in the UN study and “less than 100” in the Indian study. This is an important limitation that should be highlighted in the Abstract (which otherwise implies that the results are based on good data).

Response: We have modified Abstract to better reflect the limited data:

Results: In total, 15,547 citations were screened with two eligible studies identified that focussed on dexamethasone quality (no studies of betamethasone were identified). One study included 19 samples from 9 LMICs, and the other included “less than 100 samples” from India. The prevalence of failed dexamethasone samples ranged from 3.14% to 32.2% due to inadequate Active Pharmacological Ingredient. A higher prevalence of failed dexamethasone samples were seen at the point of care and the public sector.

Conclusions: Poor quality maternal and newborn health medicines can endanger women and newborns. Though available evidence on antenatal corticosteroids quality in LMICs is limited, results suggested poor quality dexamethasone may be prevalent in some countries. More primary studies are required to confirm these findings and guide policymakers on procurement of good-quality maternal and newborn health medicines.

26. Even though less than 100 samples were tested in the study from India, it is stated that samples from India were collected from 654 different districts where dexamethasone was available. Collecting samples from 654 different districts is inconsistent with only testing less than 100. Please clarify.

Response: We have revised the final sentence of para 1 under “Characteristics of included studies” to clarify: “Samples for the study in India were collected as part of a nationwide medicine quality study that covered 224 drug molecules. The samples were collected from across 654 different districts in India; the districts where dexamethasone was sample were not specified.”

27. Please clarify the statement “The countries that reported one failed sample of dexamethasone were Kenya, Madagascar, Nigeria, Tajikistan and Vietnam with Nepal having two failed samples” as this would total 7 samples but the text states that 6 samples failed.

Response: Thank you. That is correct 6 samples failed, Nepal had one failed sample this sentence is incorrect. This has been corrected in the paper under Main Outcomes, para 2:

“The countries that reported one failed sample of dexamethasone were Kenya, Madagascar, Nigeria, Tajikistan, Vietnam and Nepal.”

28. Would it be possible to report the exact API values instead on relying only in reaching a narrow range of API? These data would be most useful for clinicians as BMI varies by over 2 fold.

Response: Please see our response to #22 above.

29. Discussion - It is stated “We identified only two studies of 119 samples of dexamethasone…) but they authors have stated that the number of samples was 19 in one study and “less than 100” in the other. Please clarify.

Response: We have revised first para of the Discussion to better reflect this:

“This systematic review identified available evidence on the quality of dexamethasone or betamethasone in LMICs, a critical intervention in preterm birth management. We identified only two good-quality studies (19 samples and “less than 100” samples) of dexamethasone; no studies reported on betamethasone samples.”

30. Dexamethasone dosing is not exact so it is important to put into perspective the range of API found to determine the potential clinical significance of the findings.

Response: please see Response to #22 above.

---

## [Editor Report · Decision Letter 1]

16 Nov 2020

Assessing the quality of antenatal corticosteroids in low- and middle-income countries: a systematic review.

PONE-D-20-24549R1

Dear Dr. Vogel,

We’re pleased to inform you that your manuscript has been judged scientifically suitable for publication and will be formally accepted for publication once it meets all outstanding technical requirements.

Kind regards,

Chaisiri Angkurawaranon

Academic Editor

PLOS ONE
---

## [Editor Report · Acceptance letter]

19 Nov 2020

PONE-D-20-24549R1 

Assessing the quality of antenatal corticosteroids in low- and middle-income countries: a systematic review. 

Dear Dr. Vogel:

I'm pleased to inform you that your manuscript has been deemed suitable for publication in PLOS ONE. Congratulations! Your manuscript is now with our production department. 

Kind regards, 

on behalf of

Dr. Chaisiri Angkurawaranon 

Academic Editor

PLOS ONE